# The Influence of Body Weight on Semen Parameters in *Apis mellifera* Drones

**DOI:** 10.3390/insects13121141

**Published:** 2022-12-11

**Authors:** Ioan Cristian Bratu, Violeta Igna, Eliza Simiz, Ioan Bănățean Dunea, Silvia Pătruică

**Affiliations:** 1Faculty of Bioengineering of Animal Resources, University of Life Sciences “King Mihai I” from Timisoara, Calea Aradului No. 119, 300645 Timisoara, Romania; 2Faculty of Veterinary Medicine, University of Life Sciences “King Mihai I” from Timisoara, Calea Aradului No. 119, 300645 Timisoara, Romania; 3Faculty of Agriculture, University of Life Sciences “King Mihai I” from Timisoara, Calea Aradului No. 119, 300645 Timisoara, Romania

**Keywords:** drone semen parameters, drone weight, *Apis mellifera*

## Abstract

**Simple Summary:**

Knowledge and management of parental forms in bees as in other species is one of the important tools of selection and/or breeding programs. The study of honey bee drone semen may be relevant in relation to the productivity of honey bee colonies. Today, more than ever, we realize the importance of pollinators and their fine balance with the environment. Despite the importance of this aspect, currently we have less knowledge on this subject compared to other species. An adequate investigation and evaluation of the quality of the semen in relation to body weight could open new opportunities regarding its integrity, functionality and conservation in order to increase economic efficiency and maintain biodiversity in optimal conditions.

**Abstract:**

The quantity and quality of the honey bee drone semen have a significant determination on the performance of bee colonies. The existence of a smaller number of mature drones to participate in the mating of queens, as well as a sufficient number of drones but with poor quality semen can have serious implications for the productivity of bee colonies. Our study aimed to investigate the correlation between two body weight ranges of drones and semen parameters in the Buckfast honey bee, data that could be integrated into the optimization of instrumental insemination in been queens. Semen was collected from two groups of drones with different body weights (200–240 mg and 240–280 mg). Semen volume, semen concentration, motility, morphology and membrane integrity of spermatozoa were analyzed. The phenotype indicator related to body weight in correlation with the main semen parameters studied gives a weak influence or causality ratio. In drones with 240–280 mg body weight, a higher percentage of spermatozoa with abnormal morphology (>9.60%) was recorded, compared to drones with 200–240 mg body weight. The study reveals that a higher weight of honey bee drones is correlated with higher sperm concentration and total number of spermatozoa/ejaculate, with an increase in the percentage of spermatozoa with abnormal morphology.

## 1. Introduction

Honey bees are among the most important pollinators for natural and agricultural ecosystems, with an estimated 70% of agricultural crops dependent on their pollination [1,2,3,4]. In addition to their major role as pollinators, honey bees provide bee products of particular importance in human and animal therapy [5,6,7,8].

The decline of honey bee colonies, caused by many factors [9,10,11,12,13], combined with the effect of climate change on bees [14] have negative implications on harvests and implicitly on the quantity and quality of food due to the lack of pollinators [2,15] It is necessary to adapt the breeding technologies for bee colonies according to the current conditions, and to apply an effective bee reproduction program for a better management of bees.

An important factor in the natural and artificial reproduction of this species is the quality of the semen, which can be correlated with the survival and productivity of the colony [16,17]. Drone sperm quality can be influenced by genetic factors, age and body weight, seasonal variations, temperature, feeding, disease, pesticides, and storage and handling conditions [17].

In bees, natural mating is represented by copulation of a queen with an average of 12–14 drones [18], but the number of drones can be much higher [19]. Mating takes place only during flight, on clear, windless days, at lunchtime, about 3 km away from the hive and at height of 5–40 m from the ground [20] at a drone congregation area. After ejaculation, the drone’s endophallus remains trapped in the needle chamber of the queen, acting as a vaginal plug that prevents sperm from being expelled [17], and when the partners separate, it breaks, causing the drone’s death [21]. Only 3–5% of the sperm of each drone reaches the queen spermatheca and is used in the process of fertilizing the eggs [20,21].

The fertility and thus longevity, of the queen is conditioned by the quantity and quality of the semen stored in the spermatheca [22]. Tarpy et al. [23], showed that the queen can store 4–6 million spermatozoa in the spermatheca, the number required to fertilize an egg being two spermatozoa in the case of natural mating [24]. Queens mated with a small number of drones produce reproductively poor bee colonies compared to those mated with a large number of drones [25]. Queens inseminated with sperm from drones exposed to pesticides, such as Fipronil, may also be reproductively impaired due to poor semen quality [26].

The number of drones raised by a bee colony and their quality are influenced by season, food and colony strength [27]. The maximum number is reached in June and July [28], being correlated of course with the climate and the growing area. The reproductive quality of drones is determined by the quantity and quality of semen indicators such as ejaculate volume, and the number, the concentration, the viability and the length of spermatozoa [29].

According to some authors, the reproductive potential of drones is associated with body size and weight, and these are correlated with the size of the cells in which they were raised. The authors observed that drones with a higher hatching weight reached sexual maturity at 15 days and had a longer lifespan compared to drones that had a lower hatching weight [22].

Couvillon et al. [30] reported that the body weight of drones affects their mating success, with heavier drones performing an even flight throughout the day. Slone et al. [31], have shown that the ratio of drone thorax weight to body weight can affect flight ability and mating success. In addition to mating behavior, sperm quality is a key factor influencing successful reproduction in bee colonies [32].

Our research analyzed the influence of the body weight of drones on the quality of semen in *A. mellifera*. These data could be useful in optimizing the instrumental insemination of queens. We investigated the following parameters: the semen volume collected per drone, semen concentration, spermatozoa count per ejaculate collected from each drone, spermatozoa motility, morphology and membrane functional integrity. We aimed to provide additional information regarding semen quality evaluation of the honeybee drones.

## 2. Materials and Methods

In this study, we used drones of *Apis mellifera Buckfast*, mountain variety, from the central area of Romania (Sibiu).

The experiment was conducted during July–September 2021. The drones were bred between 6–30 July. At 20 days of age, the drones were transferred to the laboratory for semen collection and evaluation. In early spring 2021, bee colonies were treated against Varroosis with Amitraz, repeated after 7 days. At the beginning of June, after the extraction of acacia honey, the bee colonies were treated with a product based on essential oils. The treatments performed ensured maximum efficiency. The honey bee drones were raised in bee colonies by the method of isolating the drone frames, with the help of the Hanneman grid. The colonies of bees that raised and donated drones were housed in Dadant type hives with frame dimensions of 430 × 300 mm and of a telescope-type (430 × 150 mm), this model being the most used in Romania. All of the drone frame donor and breeder hives had two-year-old queens. Colonies were fed throughout the experiment with 1:1 (*w*:*w*) sugar syrup and protein substitutes, when due to unfavorable weather conditions nectar and pollen sources could not be properly utilized. The colonies were located in areas without melliferous agricultural plants, to prevent the possibility of contamination with insecticides from the neonicotinoid category. The maintenance and care of the drones during development and maturation are directly correlated with the size and health of the bee colonies, and the growth and preservation of the drones [33,34,35] Thus, we tried to achieve optimal conditions throughout the experiment.

Drone donor queens were arranged in isolating-type drone frames (1 frame/bee colony). After 3–4 days, the queens were released simultaneously with the assessment of the number of drone eggs laid. With the help of the markings made on the frames with the brood of drones, which indicated the predominant age of the brood, they were divided and arranged in keeper families.

One-hundred and twenty normally developed drones that had a body weight between 200–280 mg [36,37] were randomly collected from six bee colonies and brought to the laboratory in the flight box. Following the phenotypic indicator body weight, as a preliminary selection method, the drones were weighed individually and divided into two groups; each group consisted of 60 honey bee drones. Before weighing, the drones were anesthetized with CO_2_ until they became immobilized and then weighed using an analytical balance (Kern, ADJ-600-C3, Kern & Sohn GmbH, Balingen Germany, with a precision of 0.1 mg). The first group (G1) was made up of drones with recorded body weights in the range of 200–240 mg, and the second group (G2) had weights between 240 and 280 mg. The drones from the two groups were placed in wooden boxes (44 × 30 × 15 cm; L × w × h) prepared to maintain the males in optimal conditions both during transport and during the experiment in the laboratory. Thus, two cavities were placed in each box, in which 250–300 g of nurse bees and 150 g of APIFonda product (Sudzucker, Mannheim, Germany) were placed. This approach was aimed at ensuring the temperature and feeding of the drones [37,38].

The two groups of drones were transported to the Small Animal Reproduction Laboratory within the University of Life Sciences “King Mihai I” from Timisoara, where the stages of semen collection and the evaluation of its quality were carried out.

### 2.1. Collection and Determination of the Collected Semen Volume

Semen was obtained by inducing ejaculation using the partial and total endophallus eversion procedure described by Cobey et al. [39]. Carrying out a rolling and constant pressure on the two flanks of the thorax and abdomen causes the eversion of the endophallus and the exposure of the semen at the tip of the genital organ, which is generally compact, of a small spherical formation, white in color (Figure 1a), and sometimes dispersed, with an amorphous appearance (Figure 1b). Drones that did not egress properly were removed. The Schley insemination instruments (Schley Instrumental Insemination Equipment, Lich, Germany) were used for semen sampling and a large capacity ‘Wingler’ syringe (capillary 160 microliters) The sperm was aspirated into the syringe and immediately the volume of sperm collected was determined by direct reading on the graduation of the syringe. The harvesting process involved the aspiration of the entire semen. For the relevance of the study, only those drones were retained in which sperm collection could be performed for approximately 80–90% of the ejaculate. Sixty semen samples were included in the study, thirty samples per group, Semen from each male was collected separately and analyzed.

### 2.2. Evaluation of Spermatozoa Motility

From each semen sample collected from drones, a known volume of sperm (0.3 μL) was taken, which was placed in a 1.5 mL Eppendorf tube over ApiPlus extender (Minitube, Tiefenbach, Germany), which comprised the medium for the dilution and preservation of drone sperm. The sperm dilution rate was 1:4000. The mixture was homogenized using the vortex (BR-2000 Bio Rad Laboratories, Hercules, CA, USA), at 2000 revolutions/min, for 10 s.

After homogenization, 3 μL was taken from the diluted semen sample and placed in a compartment of Leja slide (Leja Products B.V., Nieuw-Vennep, The Netherlands), which was deposited on the heated stage of microscope.

Spermatozoa motility was assessed by phase contrast microscopy. The total motility (TM) of each sample was determined by visual microscopic analysis, expressed as the percentage of spermatozoa showing movement. All spermatozoa that showed motility, regardless of the form of movement, were taken into account. The motility results were expressed as a percentage. All laboratory materials used (microscope slides, pipette tips, microtubes, diluent) were maintained, until use, at a temperature of 37 °C.

### 2.3. Determination of Semen Concentration

Semen concentration (the number of spermatozoa per unit volume of semen) was determined by the hemocytometric method, using the Bürker–Türk hemocytometer (BRAND GMBH + CO. KG, Wertheim, Germany). A cover glass was applied to the hemocytometer, under which 10 μL of the diluted sperm sample with ApiPlus extender (Minitube, 151 Tiefenbach, Germany (1:4000)) was aspirated and placed into the groove of the hemocytometer. By capillary action, the fluid passed into the counting chamber. On the surface of the cover glass, corresponding to the counting chamber, a heated plate (Hamilton Thorne Bioscience, Beverly, MA, USA) was applied, which was maintained for 3 s, with the aim of stopping the movement of spermatozoa through thermal shock, thus facilitating their counting. The analysis was carried out by bright field microscopy, initially using the ×10 objective, to identify the counting grid, after which the sperm count was carried out with the ×20 objective. 

Sperm from 5 middle squares of the grid were counted, as follows: 4 middle squares located on the diagonal +1 middle square, placed anywhere on the grid. The determination of the concentration of the sperm sample was made by calculating the number of spermatozoa per μL of sperm, applying the following formula:Concentration = No. of counted spermatozoa × Dilution factor × Multiplication factor

### 2.4. Morphological Evaluation of Spermatozoa

The morphological analysis of drone spermatozoa was carried out by examining them on stained smears. Prior to the morphological study, three staining methods were tested on several semen samples: Eosin G 2% (Minitube, Tiefenbach, Germany), Spermac (Minitübe, Tiefenbach, Germany), and Diff-Quik (Medion Diagnostics, AG, Düdingen, Switzerland) [40], with the aim of choosing the optimal staining method for morphological and morphometric studies of drone spermatozoa.

The staining technique used for each stain is shown in Appendix A.

The morphological studies of the spermatozoa were carried out on semen smears, stained with Diff-Quik, the staining that provides the best observation of the structures and cellular outline of the spermatozoa, and is considered to be the optimal staining after testing of the three previously described stains. Stained smears were examined under bright field microscopy (Olympus BX51, Olympus, Tokyo, Japan) using ×40 and ×100 objectives. Two-hundred spermatozoa were evaluated for each sample, which were classified into two groups: spermatozoa with normal morphology and spermatozoa with morphological abnormalities. The percentages of spermatozoa with normal morphology and those with morphological abnormalities were calculated and the abnormalities observed in each sample were noted.

### 2.5. Morphometric Analysis of Spermatozoa

The morphometric studies were carried out on semen smears, stained with Diff-Quik staining that provided the best visualization of the sperm cell and its components in the preliminary study. The morphometric analysis was carried out by examining the spermatozoa under a light field microscope (Olympus BX51, Tokyo, Japan), using 400× and 1000× magnification, measuring the following morphological parameters: the length of the spermatozoon head, the length of the acrosome, the length of the tail and the total length of the spermatozoon. The microscopic images were taken by the digital microscopy camera (Color View II Olympus, Olympus, Tokyo, Japan). The Cell F Imaging Software for Life Science Microscopy Olympus (Soft Imaging Solution, Munster, Germany) was used to measure the spermatozoa dimensions.

### 2.6. Evaluation of the Functional Integrity of the Spermatozoa Membrane

The functional integrity of the sperm membrane was assessed using the hypoosmotic swelling test (HOST). The test consisted of placing 0.3 μL of semen in 600 μL of hypoosmotic media heated to 32 °C and homogenization, followed by keeping the sample at room temperature for 1–2 min. The microscopic examination of the samples was then carried out, in bright field, with the objective ×40, observing 200 spermatozoa and counting the spermatozoa that showed different degrees of swelling (positive reaction), these being spermatozoa with functionally intact membranes and not morphologically modified (negative reaction) present in some microscopic fields. The percentage of spermatozoa showing different forms of tail swelling was calculated for each sample. This test is an adaptation for drone spermatozoa of the test developed of Jeyendran et al. (1984) [41] to evaluate the functional integrity of the membranes of human spermatozoa.

### 2.7. Data Collection and Statistical Processing

The recorded data were statistically processed using IBM-SPSS 22.0 Software. Data are represented as means and standard deviations. Significance was assigned at a *p*-value  <  0.001, 0.01 and 0.05, with a *t*-test. Pearson correlation coefficients were calculated for each group separately.

## 3. Results

### 3.1. Semen Volume and Spermatozoa Count

The average volume of semen collected from the drones of the two experimental groups had close values: 0.7 ± 0.12 µL (range between 0.50–0.90 µL) for G1 (200–240 g) and 0.7 ± 0.10 (range between 0.50–0.85 µL) for G2 (240–280 g). There was no significant difference in semen volume between two groups (Table 1).

Semen concentration (the number of spermatozoa per microliter of semen) in G1 was 8.38 × 10^6^ spermatozoa/µL, and in G2 it was 9.16 × 10^6^ spermatozoa/µL. Comparing the results of semen concentration (Table 1), the values we obtained revealed a slightly higher concentration of sperm collected from G2, which is not statistically significant (*p* > 0.05).

The calculation of the total number of spermatozoa per ejaculate showed an average value of 5.73 × 10^6^ spermatozoa for G1 and 6.25 × 10^6^ for G2. The two groups of drones did not show significant differences (Table 1).

### 3.2. Spermatozoa Motility, Morphology and Membrane Integrity

The total motility of spermatozoa on the semen of honey bee drones from G1 ranged between 70% and 98% with an average of 88%, and that of drones from G2 ranged between 78% and 98%, with an average of 91%. G2 registered slightly higher values of the total motility than G1 (91% vs. 88%); the differences were not statistically significant (*p* > 0.05).

The average percentage of spermatozoa (Table 1) with this normal morphology was 85.30 ± 8.38% (ranging between 69–95%) in G1 and 75.70 ± 11.84% (ranging between 60–93%) in G2 (*p* < 0.05). In the case of honey bee drones with a body weight of 240–280 mg, the number of abnormal morphology spermatozoa was higher by 9.60% compared to those whose body weight was in the range of 200–240 mg.

The staining of spermatozoa with the three stains selected for the preliminary study highlighted the Diff-Quik as the optimal stain for carrying out morphological and morphometric studies. Compared to the other two stains (Eosin and Spermac), Diff-Quik provides a better visualization of the structural elements of the spermatozoa, especially the head components: the nucleus and the acrosome. In addition, the enlarged spherical region of the acrosome can be observed.

The microscopic examination reveals the filamentous aspect of drone spermatozoa and their components: the head (consisting of the nucleus and acrosome) and the tail, also named flagellum. The nucleus is narrow and linear, and continues with the acrosome, a thin and straight filament, having almost the same length as the nucleus. The other part of the nucleus continues with a long tail (Figure 2).

The morphological analysis of the drone spermatozoa revealed the presence of various abnormalities of the head (including the acrosome) and tail of the spermatozoa. The abnormal morphology of the spermatozoa head (Figure 3a–c) included the nucleus shape, size (macrocephalic, microcephalic) and number (e.g., multiple heads). Changes in the acrosome that have been identified on the semen smear were a flexed acrosome, curved acrosome and its absence (Figure 3d–f). The most common and easy to visualize changes in the microscopic field are the in morphological changes of the sperm tail: we observed bent, coiled, fragmented, short (Figure 4) and multiple tails (Figure 3c).

The results of the morphometric analysis of drone spermatozoa show large individual differences between the spermatozoa dimensions in each group (Table 2). However, between the two groups the differences are not statistically significant (*p* > 0.05), except for the head length.

The measurements performed on the spermatozoa from the G1 group showed a large variation in the length of the spermatozoa. between 156.71 µm and 260.21 µm, with an average of 230.10 ± 25.68 µm. The average length of the sperm head was 8.74 ± 0.64 µm, this representing the cumulative dimensions of the two components: the nucleus (4.72 ± 0.52µm) and the acrosome (4.02 ± 0.39 µm). The tail dimensions in G1 range from 147.73 µm to 250.90 µm with an average of 221.36 ± 25.62 µm. The average length of spermatozoa from G2 was 238.10 ± 15.17 µm, with a head length of 8.97 ± 0.27µm and tail length of 229.13 ± 15.15 µm. The average length of the nucleus was 4.84 ± 0.17 µm and that of the acrosome was 4.12 ± 0.21 µm. The dimensions obtained for the honey bee drones in G2 were higher than those in G1, for all the measured structures (Table 2) The differences were not significant (*p* > 0.05) except for the length of the head, for which we recorded significant differences (*p* < 0.05).

The results of the correlations between the dimensions of the sperm cell components measured in the present study are presented in Table 3 and Table 4. The length of the head is positively correlated (*p* < 0.05) with the length of the acrosome and with the length of the nucleus, a situation found in both groups of drones. A strong positive correlation (r = 1.00, *p* < 0.001) was identified between tail length and sperm length, regardless of the body weight of the drones.

Spermatozoa placed in a hypoosmotic environment (distilled water) immediately reacted to hypoosmolality by changing their normal morphology, especially by curling the tail. The pattern of the most observed morphological change was strongly coiled spermatozoa (Figure 5). In the classic hypoosmotic swelling test (HOST), this property is used to characterize membrane integrity [41]. In our study, this test revealed good percentages of sperm cells having a normally functioning membrane, given that the average percentage of spermatozoa with HOST positive reaction was 89% for both groups, with percentages ranging from 70% to 99% in G1 and from 74% to 98% in G2 (Table 1). An interesting aspect observed during HOST, using distilled water as a hypoosmotic medium, was the fast return of strongly coiled spermatozoa to their normal morphological shape, approximately 4–5 min after placing the spermatozoa in the osmotic medium.

## 4. Discussion

The purpose of our study was to compare some semen parameters: semen volume, spermatozoa count, motility, morphology, dimensions and functional membrane integrity of spermatozoa from honey bee drones of two range of body weight (200–240 mg and 240–280 mg). The results regarding the influence of body weight on semen quality of honey bee drones were somewhat unexpected, considering that the influence of body weight on some sperm parameters has been demonstrated. Gençer and Kahia 2011 [42] reported that in *A. mellifera* large drones weighing 221.6 mg, the mean volume of ejaculate, sperm number in drone and sperm concentration were higher than that of small drone weighing 147.3 g. On the other hand, our study differs from the one cited above by the ranges of body weight and by the origin of the drones, which could also lead to other differences between drones, not only in body weight.

Semen collection by the endophallus eversion procedure had a success rate of 70% in our study. This indicates a good result compared to other reports of bee drones of similar age. Collins and Pettis 2001 [43] stated that “artificial drone ejaculation is not completely effective”, given that only 40% of drones produced semen using the manual eversion procedure.

In our study, the two groups of drones with different body weights had similar mean ejaculate volume values: 0.7 ± 0.121 µL for G1 (200–240 mg) and 0.7 ± 0.099 for G2 (240–280 mg). This indicates that the drone body weight, on the intervals that we have selected (200–240 mg and 240–280 mg), does not influence the volume of the ejaculate. The average values of the ejaculate volume of *A. mellifera* honey bee drones, collected by eversion of the endophallus, show differences in the reports of different studies. Rhodes et al. 2011 [44] determined the mean volume of ejaculate produced by drones as 1.09 µL (range 0.72 (±0.04)−1.12 (±0.04) µL), while Woyke 1962 [45] reported that a sexually mature drone produced 1.50–1.75 µL semen. Vasfi Gençer and Kahya 2011 [42] showed that the volume of drones’ semen ranged from 0.4 µL to 1.2 µL and demonstrated that the volume of ejaculate depends on the body weight of drones. Thus, the small drones (147.3 mg) produced less semen (0.66 µL) compared to the large drones (221.6 mg) which produced 53% more semen (1.01 µL). The average ejaculate volume obtained in our study is similar to that obtained by Abdelkader et al., 2014 [38] (0.76 ± 0.17 µL), who pooled semen from different drones and then calculated an average ejaculate volume per drone. Although the average volume of ejaculate is almost equal in the two groups, in the drones with a higher body weight (240–280 mg) there is an 9.3% higher percentage for the semen concentration and 9.1% for the total number of spermatozoa per ejaculate. The season shows an obvious effect on the semen volume [46]. Additionally, a reduction of semen volume with drone age was observed [44]. The semen volume of the drones is important in practical instrumental insemination of queens, where it has been demonstrated that ejaculate volume of a drone had a significant effect on the patriline frequency; drones with larger ejaculates were consistently overrepresented in offspring [47].

Semen concentration is one of the most frequently studied parameters of semen quality in drones, allowing the spermatogenesis process and the sexual maturity of the drones to be evaluated [44]. It seems that the concentration of sperm in the ejaculate affects the reproductive success of drones, and evolution forces drones to produce not only plentiful but also particularly concentrated semen [29]. Spermatozoa is regarded as a basis for understanding several aspects of the biology of honey bee mating, including drone fitness, polyandry and sperm competition [48]. The large differences in semen concentration reported in various studies can be attributed to factors affecting this parameter, such as the technique of semen colection, the method used for sperm concentration assessment [17], the season of semen collection, age of the male [44], and drone body size (wing length as size indicator) [18]. In our study, although a higher average concentration was recorded in the group of drones with a higher body weight, the difference is not statistically significant (*p* > 0.05). The average concentration values recorded in both groups of drones (8.38 × 10^6^ spermatozoa/μL and 9.16 × 10^6^ spermatozoa/μL) are higher than those recorded in the studies of Rousseau et al. [49] (1.80 ± 1.65 × 10^6^) in drones obtained from honey bee colonies with open-mated queens belonging to two different lines, hybrid Italian stock and Buckfast stock, as well as being higher than values reported by Abdelkader et al. [38] (2.2 ± 0.6 × 10^6^ spermatozoa/μL of semen) in *Apis mellifera* L, but are similar with those reported by Collins and Pettis 2001, cited by Rhodes 2008 (9.15, range 0.5–29.25 × 10^6^ spermatozoa/μL per μL) [50]. We collected the drone semen in September, which could explain the higher values compared to other studies. Autumn drones produced higher numbers of sperm than summer drones, which produced higher amounts than spring drones, indicating a seasonal variation in sperm concentration in semen [50]. It is difficult to compare the results obtained in the analysis of sperm concentration of *A. mellifera* drones reported in different studies, both due to the relatively small number of studies [38,42] that assay this sperm parameter and due to the variations of the different factors that can influence semen concentration. Additionally, sometimes the terminology is used heterogeneously, so that the number of spermatozoa per ejaculate or per drone is reported as concentration, which actually means the number of spermatozoa per unit volume of sperm (usually microliter in drones).

Two semen collection techniques are usually used in different drone sperm quality evaluation studies: by dissecting the seminal vesicles of the drones, and by inducing ejaculation through the endophallus eversion procedure. The mean number of sperm collected per drone ejaculate is usually lower than that described for the seminal vesicles [17]. The average number of spermatozoa per drone ejaculate in our study was 5.73 × 10^6^ spermatozoa/drone ejaculate for G1 and 6.25 × 10^6^ spermatozoa/drone ejaculate for G2, both values being lower than spermatozoa numbers reported in some studies, where values ranged from 8.66 × 10^6^ to 12 × 10^6^ [18,43,45,51]. The number of spermatozoa per ejaculate ranged from 1.60 × 10^6^ to 11.52 × 10^6^ in our study. In general, spermatozoa numbers of individual *A. mellifera* drones show high variance. Several explanations for this variation were formulated, such as some errors in the application of the sperm collection method and of sample preparation for spermatozoa counting or the number of samples per drone. However, variation in sperm count is not solely due to experimental error, because large variances between drones occur even when sperms are counted by the same person [48]. Also, the sperm number produced per drone may vary according to body size, age, season, genetics and diseases [18,44,49].

The motility of honey bee spermatozoa has been evaluated only occasionally [52] in contrast to mammals where sperm motility is one of the most widely used parameters to determine sperm quality [17]. However, sperm motility is a prerequisite for sperm migration to the queen’s spermatheca and for subsequent egg fertilization. It has been shown that sperm motility is more strongly correlated with sperm performance indicators in inseminated queens than other sperm quality tests, including the viability assay [53].

Spermatozoa of *A. mellifera* honey bee drones of an Italian strain are long and filamentous cells with a length of 250–270 µm. The head of a honey bee sperm is an elongated structure that consists of a nucleus (6.6 µm) and an acrosomal complex, (5.6 µm). The entire head region measures 12.2 μm in length [54]. The morphometric studies carried out on mature Carniolan drones at the beginning and end of the mating season showed that the dimensions of drone spermatozoa from June (273.50 ± 16.58 µm) were somewhat longer than spermatozoa from August (257.97± 16.37 µm), and the individual elements of their structure were longer as well [55]. In our study, the average length of spermatozoa (230.10 ± 25.68 µm/G1, 238.10 ± 15.17 µm/G2) was smaller, in both groups. Additionally, the length of the head and its components, the nucleus and the acrosome, were smaller. Data reported by Tarliyah et al. [56] in *Apis Mellifera* L., indicate shorter honey bee drone spermatozoa than those of our study. The average length of the sperm cell was 217.57 µm [56]. It is difficult to explain these differences, due to variables from different studies, which could influence spermatozoa size, such as the variety/strain of drone, body size, method of sperm collection (seminal vesicle section, ejaculation induction), the season of sperm collection and measuring method. It is possible that the season in which the spermatozoa from our study were collected (September, the end of mating season) influences the length of the sperm cell. According to Gomendio and Roldan [57], longer sperm cells in mammals may be an adaptation that increases their competitiveness. This also applies to the sperm of birds [58]. The study on the influence of sperm competition on sperm design in mammals using a large data set (226 species) shows that sperm competition is associated with an increase in total sperm length, which results from an increase in size of all sperm components: head, midpiece, principal plus terminal piece and, hence, flagellum. The increase in sperm length was found to be associated with enhanced swimming velocity, an adaptive trait under sperm competition [59].

In the case of drones with a lower body weight (G1/200–240 mg), the percentage of morphologically normal spermatozoa was significantly higher (11.48%) than in drones with a higher body weight (G2/240–280 mg). Comparing the results of the morphological analysis with the other sperm parameters, it was observed that the increase in the concentration and total number of spermatozoa in the ejaculate is associated with the increase in the percentage of abnormal spermatozoa.

The unique morphological characteristics of spermatozoa are critical to the functionality of the cell. Different factors can negatively affect the production of normal sperm. Many spermatozoa abnormalities have been documented to be associated with male infertility and sterility in most studied species [60]. Very few studies have assessed the presence of morphologically abnormal spermatozoa in honey bee drone semen [17]. Different aberrant tail forms, including coiled, frayed and double-ended forms, have occasionally been described in drones [55,61]. The choice of the three stains for the morphology and morphometry in bright field microscopy for our study was based on their routine use for the spermatozoa of different animals and humans and also on the fact that we identified few stains used for the analysis of honeybee drone spermatozoa. We also took into account the observation that the stain and staining procedures used in humans or domestic animals are not always applicable to other species [60]. To our knowledge, the only study was performed by Gontarz et al. [55] who investigated morphology and morphometry of drone spermatozoa using staining techniques and examination under bright field microscopy. They used two stains: a complex of eosin and gentian violet according to a method described by Kondracki et al. [62] and silver nitrate AgNO_3_ in a colloidal gelatin solution according to Andraszek and Smalec [63]. Both staining methods allow the identification of individual spermatozoa structures, however, the eosin + gentian violet complex stain the drone spermatozoa very well, while spermatozoa stained with silver nitrate AgNO_3_ in a colloidal gelatin solution are somewhat less distinct [55]. In morphology and morphometry studies of drone spermatozoa it is useful to stain drone sperm cells using various techniques, because each technique can reveal different details of the sperm ultrastructure or defects in its morphological structure [64]. The results of our study show that all three stains used in the preliminary study to choose the optimal stain for morphology and morphometric analysis can be used to stain honeybee drone spermatozoa. Diff-Quik staining better identified the acrosome and a structural component, namely the enlarged spherical region of the acrosome. This acrosome structure was identified in transmission electron microscopy, following fixation of the sperm by high-pressure freezing and freezing substitution [54]. The acrosomal complex of honey bee drone sperm consists of an anterior tubular acicular apex, an enlarged spherical region, and the elongated acrosomal proper. Internally, the structures of acrosomal complex confirm the general description of the tri-layer model: an extra-acrosomal layer, an acrosome vesicle, and a central acrosomal rod located in a ubacrosomal cavity. The acrosomal rod terminates anteriorly with an electron dense corpuscle in an ensheathing cap at the spherical region. The electron micrographs also show the presence of a centriolar adjunct and a structure possibly of a centriolar or basal body origin [54].

Membrane integrity is not only important for sperm metabolism, but a correct change in the properties of the membrane is required for successful union of the male and female gametes. Thus, the integrity and functional activity of the sperm membrane is of fundamental importance in the fertilization process, and assessment of membrane function may be a useful indicator of the fertilizing ability of spermatozoa [41]. The hypoosmotic swelling test (HOST) was used to evaluate the functional integrity of the sperm membrane for the spermatozoa of many species of domestic animals and humans, as well as honey bee drones [41,65,66,67]. When exposed to hypo-osmotic conditions, water will enter the spermatozoon in an attempt to reach osmotic equilibrium. This inflow of water will increase sperm volume and the plasma membrane will bulge (balloon). This ballooning effect will be referred to as “swelling” [41]. Tail swelling from the exposure of spermatozoa with intact plasma membranes to hypo-osmotic solution indicates that the transport of water across the membranes is occurring normally and the functional integrity of the membrane has been preserved [41]. Swollen spermatozoa are identified by changes in the shape of the cell, as indicated by coiling of the tail and are considered as intact [68]. In our study a high percentage (89%) of sperm cells showing good behavior of their membrane in face of osmotic stress (hypoosmotic medium) was observed in both groups. It seems that this sperm parameter is not affected by the body weight of the honey bee drone.

## 5. Conclusions

The results of our study show that most of the sperm parameters are not influenced by the body weight of the honey bee drones. A significantly higher percentage of spermatozoa with normal morphology was observed in drones with smaller body weight (200–240 mg versus 240–280 mg). In conclusion, a higher weight of honey bee drones is related with a higher concentration and total number of spermatozoa/ejaculate, but also with an increase in the percentage of spermatozoa with abnormal morphology.

## Figures and Tables

**Figure 1 insects-13-01141-f001:**
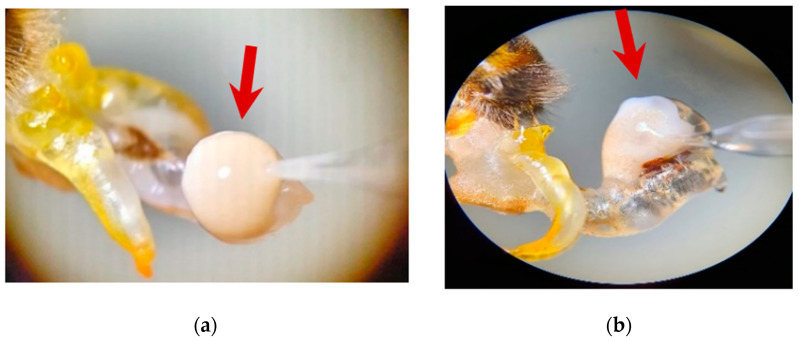
Semen collection from (**a**) spherical ejaculate or (**b**) dispersed, amorphous ejaculate.

**Figure 2 insects-13-01141-f002:**
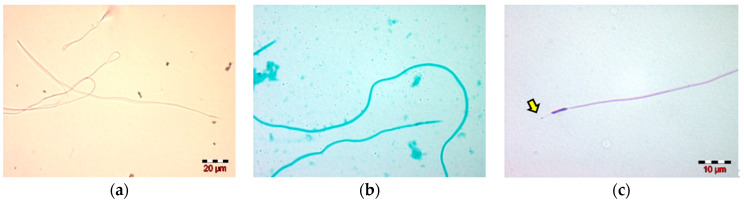
Microscopic images: (**a**) drone spermatozoa stained with Eosin G; (**b**) Spermac; (**c**) Diff-Quik.

**Figure 3 insects-13-01141-f003:**
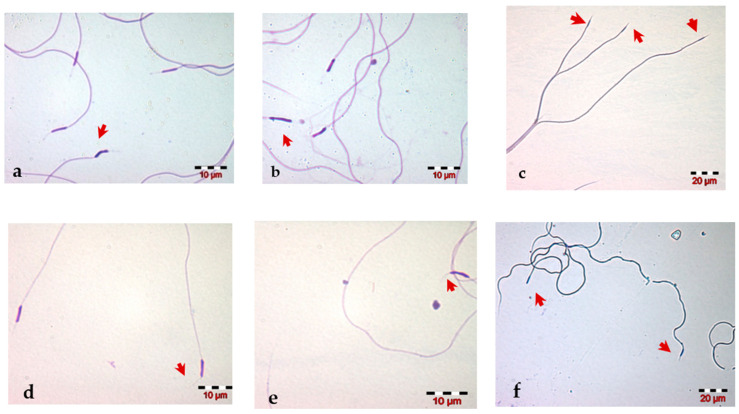
Microscopic images: abnormal morphology of (**a**–**c**) honey bee drone sperm head and (**d**–**f**) acrosome.

**Figure 4 insects-13-01141-f004:**
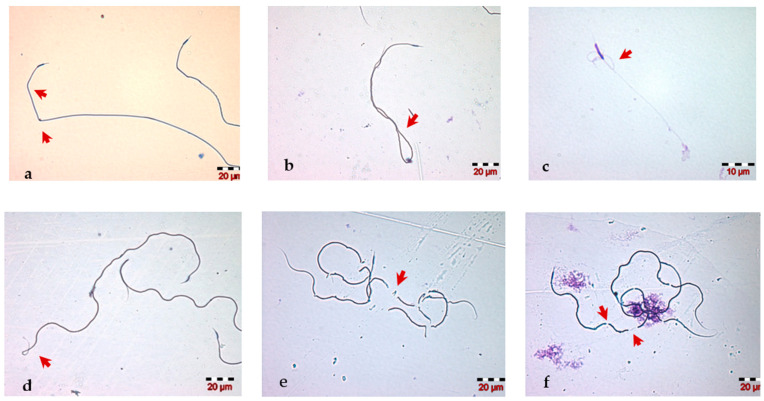
Microscopic images with abnormal morphology of spermatozoa tail: (**a**) bent, (**b**,**d**) coiled, (**c**) affected structure, (**e**,**f**) fragmented.

**Figure 5 insects-13-01141-f005:**
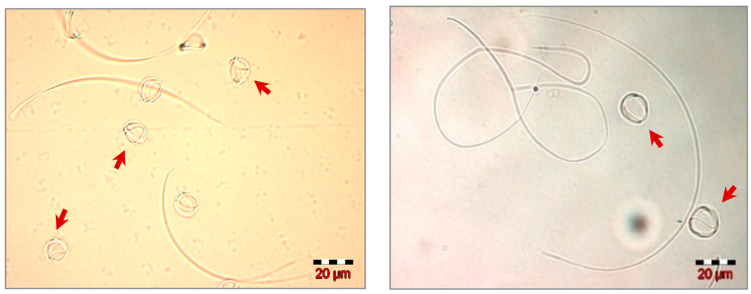
Morphological appearance of swollen (strongly coiled) honey bee drone spermatozoa.

**Table 1 insects-13-01141-t001:** Evaluation of sperm parameters in *A. mellifera* drone.

Sperm Parameters	Group	*n*	Mean	SD	Values		*p* Values
					Min	Max	
Semen volume (μL)	G1 ^1^	30	0.70 ^a^	0.12	0.50	0.90	0.991
	G2 ^2^	30	0.70 ^a^	0.10	0.50	0.85	
Semen concentration (nr. spz/μL)	G1	30	8.38 × 10^6 a^	4.09 × 10^6^	3.8 × 10^6^	14.6 × 10^6^	0.163
	G2	30	9.16 × 10^6 a^	4.54 × 10^6^	3.20 × 10^6^	14.9 × 10^6^	
Total number spermatozoa/ejaculate	G1	30	5.73 × 10^6 a^	2.72 × 10^6^	2.28 × 10^6^	11.52 × 10^6^	0.488
	G2	30	6.25 × 10^6 a^	3.00 × 10^6^	1.60 × 10^6^	9.5 × 10^6^	
Total motility (%)	G1	30	88.00 ^a^	8.13	70.00	98.00	0.481
	G2	30	91.00 ^a^	8.30	70.00	98.00	
Morphology (% of normal spermatozoa)	G1	30	85.30 ^a^	8.38	69.00	95.00	0.001
	G2	30	75.70 ^b^	11.84	60.00	93.00	
Membrane integrity (%)	G1	30	89.07 ^a^	8.16	70.00	99.00	0.787
	G2	30	89.60 ^a^	7.00	74.00	98.00	

^1^ G1 = group of drones weighing 200–240 mg; ^2^ G2 = group of drones weighing 240–280 mg. ^a-a^ *p* > 0.05, ^a-b^ *p* < 0.05.

**Table 2 insects-13-01141-t002:** Results of morphometric analysis of honey bee drone spermatozoa.

Sperm Parameters	Group	*n*	Mean	Std. Deviation	Values		*p* Values
					Min	Max	
Acrosome length (μm)	G1 ^1^	50	4.02 ^a^	0.39	3.16	4.74	0.111
	G2 ^2^	50	4.12 ^a^	0.21	3.37	4.71	
Nucleus length (μm)	G1	50	4.72 ^a^	0.52	3.79	5.68	0.104
	G2	50	4.84 ^a^	0.17	4.44	5.31	
Head length (μm)	G1	50	8.74 ^a^	0.64	7.40	10.42	0.023
	G2	50	8.97 ^b^	0.27	8.00	9.75	
Tail length (μm)	G1	50	221.36 ^a^	25.62	147.73	250.90	0.068
	G2	50	229.13 ^a^	15.15	148.87	254.31	
Spermatozoa length (μm)	G1	50	230.10 ^a^	25.68	156.71	260.21	0.061
	G2	50	238.10 ^a^	15.17	158.19	263.54	

^1^ G1 = group of drones weighing 200–240 mg (*n* = 50); ^2^ G2 = group of drones weighing 240–280 mg (*n* = 50). ^a-a^ *p* > 0.05, ^a-b^ *p* < 0.05.

**Table 3 insects-13-01141-t003:** Pearson coefficients (r) showing correlation between the morphometric dimensions of drone spermatozoa (G1).

Morphometric Traits	Nucleus Length	Head Length	Tail Length	Sperm Length
Acrosome length	−0.020 (*p* > 0.05)	0.591 (*p* < 0.001)	−0.028 (*p* > 0.05)	−0.013 (*p* > 0.05)
Nucleus length	-	0.795 (*p* < 0.001)	0.131 (*p* > 0.05)	0.150 (*p* > 0.05)
Head length	-	-	0.089 (*p* > 0.05)	0.113 (*p* > 0.05)
Tail length	-	-	-	1.00 (*p* < 0.001)

**Table 4 insects-13-01141-t004:** Pearson coefficients (r) showing correlation between the morphometric dimensions of drone spermatozoa (G2).

Morphometric Traits	Nucleus Length	Head Length	Tail Length	Sperm Length
Acrosome length	−0.012 (*p* > 0.05)	0.773 (*p* < 0.001)	−0.122 (*p* > 0.05)	−0.108 (*p* > 0.05)
Nucleus length	-	0.625 (*p* < 0.001)	0.250 (*p* > 0.05)	0.260 (*p* > 0.05)
Head length	-	-	0.063 (*p* > 0.05)	0.081 (*p* > 0.05)
Tail length	-	-	-	1.00 (*p* < 0.001)

## Data Availability

The data used in this study are available by email request to the corresponding author.

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
