# Peer review of "The Influence of Body Weight on Semen Parameters in Apis mellifera Drones"

_insects, 2022, doi:10.3390/insects13121141_

Round 1

Reviewer 1 Report

The paper by Bratu et al. tries to investigate the relationship between body weight and several semen parameters in honey bee drones. The topic of the manuscript is not novel nor original and the experimental procedure presents a number of potential flaws and weaknesses that could undermine the solidity of the results.

Lines 14-21: Selection of honey bee bee lineages through specific breeding programs does not represent a novel or original topic and also the potential role of body size in sperm quality of drones has been investigated over the years (Schlüns et al. 2003; Rangel and Fisher 2019; Gençer and Kahya 2020; Yàniz et al. 2020). The authors should rephrase their summary to give an edge on the results of their study and what they add in terms of knowledge to the already available body of research on the subject.

Lines 22-24: It is not immediately clear for the reader why “quantity and quality of the honey bee drones' semen are factors with major significant determination on the performance of bee colonies”, the authors should add some information on how these traits can influence the performance of bee colonies.

Line 26: Apis mellifera in italics.

Line 29: weak

Lines 30-31: The authors might omit this sentence from the abstract.

Line 40 and following: When reporting cited literature, please include all the reference number in  single brackets. E.g. [1-3, 5, 7,  11] instead of [1], [2], [3], [5], [7], [11].

Line 42: when talking about bee decline, the authors should cite other papers on old and recent drivers of this decline such as:

-       Goulson, D., Nicholls, E., Botías, C., & Rotheray, E. L. (2015). Bee declines driven by combined stress from parasites, pesticides, and lack of flowers. Science, 347(6229), 1255957.

-       Eliash, N., & Mikheyev, A. (2020). Varroa mite evolution: a neglected aspect of worldwide bee collapses?. Current Opinion in Insect Science, 39, 21-26.

-       Cappa, F., Baracchi, D., & Cervo, R. (2022). Biopesticides and insect pollinators: Detrimental effects, outdated guidelines, and future directions. Science of The Total Environment, 155714.

Lines 47-49: this is a very crucial statement on which the premises of the present work are based. The authors must provide references supporting such a statement.

Line 52: I would rather use “natural mating is represented by copulation between…” rather than “sexual intercourse”.

Line 55: to add further emphasis to the relevance of their study, the authors should mention the fact that social hymenopteran males, including honeybee drones, have always attracted little attention in terms of research, when compared to their female counterparts (both queens and workers). To point out this bias in attention and research, the authors should cite:

-       Beani, L., Dessì-Fulgheri, F., Cappa, F., & Toth, A. (2014). The trap of sex in social insects: from the female to the male perspective. Neuroscience & Biobehavioral Reviews, 46, 519-533.

-       Heinze, J. (2016). The male has done his work—the male may go. Current Opinion in Insect Science, 16, 22-27.

Line 57: “mating plug” rather than “vaginal plug”.

Line 61: avoid repeating “with which the queen has mated” already mentioned in the previous sentence.

Line 72: replace reference with number.

Lines 88-91: as already pointed out, this aim of the study does not focus on a particularly novel subject. The authors should try and stress what they expect to add in terms of relevant findings and results to the available scientific literature.

Line 95: “the experiment was carried out”.

Lines 109-110: it is not clear what the authors mean with “released simultaneously with the assessment of the weight of drone eggs laid”.

Lines 93-131: the authors do not provide any sample size. How many colonies? How many frames? How many drones in each group?

Lines 113-120: I believe here there is a potentially strong weakness in the method. Do the authors consider also other parameters (such as body size, size of the different body parts, etc.) apart from body weight to separate the two experimental groups of drones? Body weight is not a fixed parameter; for example, a drone that has recently been fed by nestmate workers might be heavier than a starved male, even if their body size or weight as starved is the same. Thus, given the small range of weight differences between the two groups (200-240 mg vs 240-280 mg) I fear that the results of the study might not be altogether solid.

Lines 119-121: provide measures and details of the boxes used and move Figure 1 to supplementary material. The representation of the experimental apparatus in picture is very old-fashioned. This is valid also for figure 2 and 3. Please remove them from the manuscript and move them to supplementary material.

Lines 144-145: how was calculated/estimated the 80-90% of the ejaculate?

Lines 133-145: No sample size is reported. I imagine that the semen of different males from the two groups was collected in a sample. First, this might cause a bias, for example if the semen of a particular male contained a high proportion of abnormal sperm. Moreover, from how many drones of the two weight groups the semen samples were collected? The authors must provide this information.

Lines 149-163: once more, how many samples?

Paragraphs 2.4.1 -2.4.3: move this paragraph to supplementary materials.

Results and Discussion: the authors should try and clarify the potential flaws in their experimental procedure before the results and their interpretation and discussion can be considered solid.

Author Response

Dear Reviewer,

Thank you for taking the time to review our manuscript. We greatly appreciate the suggestions made to improve our paper: “The influence of body weight on semen parameters in Apis mellifera drones“.

Sincerely,

The authors.

Reviewer 2 Report

General comment

This study deals with the relationship between the body weight of drones and the quality of their semen.

This study is interesting because it explores different parameters to judge the quality of semen and spermatozoid. However, the extension in terms of beekeeping and queen insemination does not appears as important as reported by the authors because the weighing of drone during the procedure of semen collection is relatively difficult to carry out.

Major comments

L153. This homogenization procedure appeared somewhat brutal? How was it checked that it does not alter the structure et the viability of spermatozoa?

L276-287. For these parameters, because cells were counted, the authors should be able to provide data on the proportions of spz presenting enlarged spherical region of the acrosome.

Minor comments

L108. Why eggs are weighed?

Minor comments are inserted directly on the manuscript.

Editorial revisions

Editorial revisions are indicated on the text.

Author Response

Dear Reviewer,

Thank you for taking the time to review our manuscript. We greatly appreciate the suggestions made to improve our paper: “The influence of body weight on semen parameters in Apis mellifera drones“

Revisions to the manuscript were marked up using the “Track Changes” option. We indicated the current line in the working version of manuscript.

Sincerely,

The authors.

Reviewer 3 Report

Comments to Author

The manuscript has important findings. There are certain sections (MM, Results and discussion) were text needs author's attention.

1.       Add authority name with scientific name in the manuscript (where applicable)

2.       Correct the grammar of the sentences/ Rephrase the sentences for better understanding.

3.       Use abbreviation of genus name after the first use of its complete name: Apis mellifera and later A. mellifera

4.       Check space between words

Introduction

5.       At certain points in the introduction, general information could be shortened or omitted

6.       Improve the objectives of study at the end of the introduction (see detailed comments in the manuscript)

MM:             

7.       Add suitable references to support the protocols mentioned in MM

8.       How weight of drone’s eggs was measured?

9.       Figure 2 should be properly labeled to show the part of interest.

10.    Which medium for determination of sperm concentration was used and prepared?

Results

11.    Arrange the parameters in tables according to their sequence of description in the text

12.    Figure 3 should be properly labeled to show the part of interest under description in the text.

13.    It is hard to follow the description of text with figures. Because, figures are without any label/ arrow/ square to highlight the area of interest

14.    Figure 7: Add A and B for different section of figure

15.    Cross check the values with table

Discussion:

16.    Some parts of discussion are just repetition of results without any proper discussion.

17.    Clearly mention the species of honey bee with authority.

18.    At certain points (see detailed comments in the manuscript), there is a need to establish a link of current results with findings of previous studies.

19.    Certain facts are discussed which are in fact in fall in the scope of present study. (Example: Scope of present study is only the weight of drone and its effect on different attributes of semen: Line 377-379), (Present study did not investigate any seasonal effect on weight and semen: line 381-382), (The current study does not have the comparison of different season/ months: Line 405)

20.    Cross check the references and data quoted in line 401-403, where it is stated that semen concentrations found in present study was higher than previous studies

21.    The author must mention the type of subspecies of Apis mellifera used in previous studies and in current study that might be a possible reason for such variation.  If author can compare the data of same subspecies (as investigated in present study), a scenario could be better explained (Line 425-428)

22.    Cross check the values with table (Line 519)

Conclusion:

23.   Rephrase the sentences for better understanding

Author Response

(The authors gave the same response as above.)

Reviewer 4 Report

In general, very interesting study on a topic not so studied. It provides some new insights and informations. After reading a manuscript, I have a feeling that authors could maybe have better results if drones of smaller weight were also included and maybe three groups instead of two. Further, I don’t understand why the study on drones was conducted in September when usually drones are “kicked” out of colonies at this time of year. Finally, in such a study, Varroa infestation should be checked as Varroa may have a crucial effect on health of drones. Abstract need to be rewritten as it is quite unclear at the moment. Introduction needs some minor changes. In methodology part I am missing several important informations. Results and discussion also need some adaptation and changes.

More detailed comments:

Abstract

Abstract is unclear. It should provide general information on the structure of the experiment (you have two groups), main findings and conclusion. It should be made in a way that is easy to read by a reader who encounters the article for the first time.

Introduction

L22-23 – queen fertilization station – the common wording in English is mating station, not fertilization station

L26-27 – be specific here, mention you worked on Buckfast bee. This is quite important to mention in the abstract as there are differences between subspecies.

L38 – after you mention honey bees for the first time, please provide scientific Latin name (Apis mellifera L.) afterwords.

L41 – there is no need for 6 references here. And these are mostly studies from Romania. I suggest to have 3-4 more general references here.

L42 – I noticed on several places you write bee colonies. You should be specific here and always refer to as honey bee colonies.

L42 – “The decline of bee colonies, caused by many factors [12],[13]” – here you provide only two references that are decade ago while in the previous line you provide 6 references. Here you should find and cite studies that confirm this claim and talk about factors that are causing decline of honey bee colonies. Also, I am sure there are far more relevant studies on effect of climate change than [14]. I advise you to be objective and skip self-citation if it is not the best option.

L44 – can you confirm with the reference that there is a lack of pollinators?

L66-70 – how this can be in the same sentence? The first one is on a reproductive biology and mating, and the second one about insecticides. Queens mated with less drones may produce “poor” colonies, but it is not a rule per se.

L71 – yes, season and food are important, but the colony strength and development are crucial! You may have the best spring ever with plenty of available food inside and outside the hive. But if the colony is weak, there will be no drones produced!

L72 - (Rowland, 1987) – cite in a proper format, it is missing in the reference list.

L72 – “the maximum number is reached in June and July” – is this for your region? You have to be aware that in other parts of the Europe and world these months are different.

Methods

General questions regarding methodology:

-       How many drones were analysed?

-       How many colonies were used?

-       Why you decided to choose these two groups (what was the criteria)? By eliminating light and heavy drones the possibility of determining differences is reduced. Maybe you should have created three groups?

-       do you know the age of drones used for the analysis? If yes, how did you determined the age?

-       why is experiment conducted in September?

-       do you know the infestation rate of colonies with varroa mites? Varroa is a very important factor influencing most of measured parameters on drones.

L93 – Buckfast is a breed not subspecies of bee, therefore, it should not be Apis mellifera buckfast rather just Buckfast bee.

L94 – put a dot at the end of the sentence.

L95 – the experiment “was” carried out…

L95 – in September of which year?

L96- this is unclear. What does “isolating the drone frames in the shed” means? Please use standard English wording!

L99-100 – are these queens sister queens?

L100 – not families but colonies.

L104-106 – please provide reference here.

L109 – if I understood correctly here, you weighted the drone eggs, am I correct? If so, you don’t present any result about that.

L113-114 – how old were the drones collected? How did you determined the age?

L230 – how did you measured the length of the tail and length of spermatozoa?

Results

General comment – there is no need to report all the results in the text if they are reported in the table!

L253 – after ± is SD or SE? This goes for all values you present in the text because sometimes you show SD and sometimes SE.

L254 – Hm, this is little harsh conclusion. You didn’t test the effect of body weight on semen volume, you just compared volume of semen in drones of two groups. Different methodology and statistical test are needed to conclude this. It is important to show distribution (frequency) on the weight of drones from each group (200-210 xx drones, 211-220 xx drones etc. but as a simple graph). Instead of sentence you wrote, I suggest to go with something like this: there were no significant difference in sperm volume between two groups (t-test results, p-value).

L256-258 (table 1) – ne need to have Apis mellifera  in the title of table. Also, I suggest to move results of total motility after semen concentration and total number of spermatozoa.

L265 – beside p value it is always recommended to present t-test results. This goes for all p-values you present.

L300-301 – the pictures d, e and f are not marked on the photos (I suppose the three lower once from left to right).

L311 – here you present results of Mean ± SD. In the previous part of results you presented Mean ± SE. Why did you choose this? Why you present SE in this kind of descriptive analysis?

Discussion

L347 – please be aware, you didn’t measured the influence (or the effect) of body weight on these parameters, you compared these parameters in two groups of drones.

L351 – honey with a single “n” (not honney).

L368-369 – probably this is a consequence of small difference in weight between groups. When you decided to eliminate drones lighter than 200 mg you eliminated the possibility to find differences. And this you conclude very nicely in next sentence.

L374-377 – this is not important at all as there are no significant differences.

L383 -384 – please provide reference for “a reduction of semen volume with drone age was observed”.

L406-408 – hm, this is completely different to the statement in lines 380-383 (In autumn, drones produced the smallest volumes compared with spring and summer [38]. This finding could explain the lower volume of ejaculate obtained in our study, compared to the previously mentioned studies).

L406 – you should be very careful and aware that the citation 46 is reporting on Australia. Their Autumn is much different than Autumn in Romania and this could not be compared

L452 – well I would not agree that the season is affecting the length of the spermatozoa. All the references you use in next few sentences are describing an evolutionary adaptation not the seasonal variation on the sperm length.

L523-524 – you should highlight here that this is for your study for your groups because there are studies that used drones of smaller weight and they found differences.

References

I suggest a detailed review of all references

L556 – you are missing the first half of the title of this paper!

L561 – cite the reference according to the guidelines (journal is missing)

L642 – not sure that this is the proper way to write reference

Author Response

(The authors gave the same response as above.)

Round 2

Reviewer 1 Report

I acknowledge that the authors tried to answer to the comments on the previous version, but some of the answers are not entirely satisfactory. For example, the answer to the comment on the potential bias in the methodology (lines 113-120) linked to the choice of body weight as parameter to separate the two experimental groups of drones or the subjective evaluation of the ejaculate percentage. Furthermore, as already pointed out in my previous revision, I believe that the topic of the manuscript is neither original nor novel and the results of the study might be suitable for a more specific journal on honey bees or apiculture.

Author Response

(The authors gave the same response as above.)

Reviewer 4 Report

L26-27 - Buckfast is a hybrid. So you should say Buckfast honey bee not Apis mellifera Buckfast.

L28-34 - not sure you should write (2) Methods, (3) Results... in abstract. I think this is not needed, probably editor will make a suggestion here.

L119 - "collected from six strength bee colonies" - I think you should delete "strength".

L226 - I couldn't find you report standard error anywhere now, so you should delete it here.

L276 - problem with a figure 3. you added picture numbers d, f and g instead of d, e, f.

L283 - dot is missing at the end of the sentence.

Author Response

Dear Reviewer,

Thank you for taking the time to review our manuscript. We greatly appreciate the suggestions made
to improve our paper: “The influence of body weight on semen parameters in Apis mellifera drones“
We have addressed your suggestions as follows:
L26-27 - Buckfast is a hybrid. So you should say Buckfast honey bee not Apis mellifera Buckfast.
We replaced “Apis mellifera Buckfast” with “Buckfast honey bee”.
L28-34 - not sure you should write (2) Methods, (3) Results... in abstract. I think this is not
needed, probably editor will make a suggestion here.
We removed the headings of structure: (2) Methods, (3) Results and (4) Conclusions.
L119 - "collected from six strength bee colonies" - I think you should delete "strength".
We deleted "strength".
L226 - I couldn't find you report standard error anywhere now, so you should delete it here.
We deleted.
L276 - problem with a figure 3. you added picture numbers d, f and g instead of d, e, f.
We rectified (d,e,f). Thank you!
L283 - dot is missing at the end of the sentence.
We added the dot.

We are grateful for your time and effort to review our paper and we hope we have successfully
addressed all your queries!
Sincerely,
The authors.